# Anticholinesterase and Antityrosinase Secondary Metabolites from the Fungus *Xylobolus subpileatus*

**DOI:** 10.3390/molecules29010213

**Published:** 2023-12-30

**Authors:** Kristóf Felegyi, Zsófia Garádi, Elżbieta Studzińska-Sroka, Viktor Papp, Imre Boldizsár, András Dancsó, Szabolcs Béni, Przemysław Zalewski, Attila Ványolós

**Affiliations:** 1Department of Pharmacognosy, Semmelweis University, 1085 Budapest, Hungary; felegyi.kristof@pharma.semmelweis-univ.hu (K.F.); garadi.zsofia@pharma.semmelweis-univ.hu (Z.G.); boldizsar.imre@semmelweis.hu (I.B.); szabolcs.beni@ttk.elte.hu (S.B.); 2Directorate of Drug Substance Development, Egis Pharmaceuticals Plc., 1475 Budapest, Hungary; 3Department of Pharmacognosy and Biomaterials, Poznan University of Medical Sciences, Rokietnicka 3 Str., 60-806 Poznan, Poland; elastudzinska@ump.edu.pl (E.S.-S.); pzalewski@ump.edu.pl (P.Z.); 4Department of Botany, Hungarian University of Agriculture and Life Sciences, 1118 Budapest, Hungary; papp.viktor@uni-mate.hu; 5Department of Plant Anatomy, Institute of Biology, Eötvös Loránd University, 1117 Budapest, Hungary; 6Department of Analytical Chemistry, Institute of Chemistry, Eötvös Loránd University, 1117 Budapest, Hungary

**Keywords:** medicinal mushroom, anticholinesterase, antityrosinase, fomannoxin, triterpenes, *Basidiomycota*

## Abstract

*Xylobolus subpileatus* is a widely distributed crust fungus reported from all continents except Antarctica, although considered a rare species in several European countries. Profound mycochemical analysis of the methanol extract of *X. subpileatus* resulted in the isolation of seven compounds (**1**–**7**). Among them, (3β,22E)-3-methoxy-ergosta-4,6,8^14^,22-tetraene (**1**) is a new natural product, while the NMR assignment of its already known epimer (**2**) has been revised. In addition to a benzohydrofuran derivative fomannoxin (**3**), four ergostane-type triterpenes **4**–**7** were identified. The structure elucidation of the isolated metabolites was performed by one- and two-dimensional NMR and MS analysis. Compounds **2**–**7** as well as the chloroform, *n*-hexane, and methanol extracts of *X. subpileatus* were evaluated for their tyrosinase, acetylcholinesterase, and butyrylcholinesterase inhibitory properties. Among the examined compounds, only fomannoxin (**3**) displayed the antityrosinase property with 51% of inhibition, and the fungal steroids proved to be inactive. Regarding the potential acetylcholinesterase (AChE) inhibitory activity of the fungal extracts and metabolites, it was demonstrated that the chloroform extract and compounds **3**–**4** exerted noteworthy inhibitory activity, with 83.86 and 32.99%, respectively. The butyrylcholinesterase (BChE) inhibitory assay revealed that methanol and chloroform extracts, as well as compounds **3** and **4,** exerted notable activity, while the rest of the compounds proved to be only weak enzyme inhibitors. Our study represents the first report on the chemical profile of basidiome of the wild-growing *X. subpileatus*, offering a thorough study on the isolation and structure determination of the most characteristic biologically active constituents of this species.

## 1. Introduction

*Xylobolus* P. Karst. is a small, globally distributed wood-inhabiting fungal genus classified within the order *Russulales* and the family *Stereaceae* [1]. The type species of the genus is *X. frustulatus* (Pers.) P. Karst., a distinctive corticioid species that grows on dead or decaying oak wood and forms perennial basidiomata, which crack into small, angular polygons. Another species, *X. subpileatus* (Berk. & M.A. Curtis) Boidin (≡*Stereum subpileatum* Berk. & M.A. Curtis), is also associated with oaks [2] but has an effused-reflexed basidiomata resembling members of the *Stereum* genus [3]. However, unlike *Stereum* species, the basidiomata of *X. subpileatus* are perennial, and acanthocystidia can be observed in its hymenium under microscopic examination [4]. The basidiomata of *X. subpileatus* are characterized by their concentric zonation, featuring an upper surface of cinnamon or reddish-brown color, densely covered with a felty or tomentose layer. The actively growing edge of the basidiomata displays a vibrant yellow color, while the smooth or tuberculate hymenophore is cream-colored, pale ochre, later cracking, and becoming more or less grayish [5,6]. This species was described from South Carolina by Berkeley and Curtis [7], but it has since been reported from all continents except Antarctica (Global Biodiversity Information Facility, 18 June 2023). In Europe, *X. subpileatus* has a mostly southern distribution and is considered a rare species in several countries [8]. It has a limited ethnomedicinal importance, used in China [9] and Pakistan [10] for various diseases, including flu, fever, high blood pressure, and cancer. According to our search, the literature on this species is rather scarce; only few reports are available about its chemistry and biological activity. In a study by Birkinshaw et al., *X. subpileatus*, found growing in Persian-oak beer barrels and causing contamination of the beer, was cultured on two different media. The profiles of steam-volatile products of the obtained cultures were examined, and the three major compounds were found to be cinnamaldehyde, cinnamic acid, and 5-methoxycoumarone [11]. Several benzofuran derivatives were identified by Bu’Lock et al. from cultures of *X. subpileatus* [12], while Tian et al. investigated the ethyl acetate extract of the culture broth and methanol extract of the mycelium of a strain of *X. subpileatus* CGMCC5.57, which resulted in the isolation of a new dihydrobenzofuran and six other known fungal metabolites [13].

Despite the abovementioned few experiments performed on culture broth of *X. subpileatus*, our current study is the first to investigate the chemical profile of the wild-growing fruiting bodies of this species.

## 2. Results and Discussion

Thorough investigation of the methanol extract obtained from the lyophilized sporocarps of *X. subpileatus* led to the identification of seven constituents (**1**–**7**) (Figure 1). The fungal extract was first subjected to solvent–solvent partition between aqueous MeOH and *n*-hexane, followed by extraction with chloroform and ethyl-acetate. The resulting *n*-hexane and chloroform extracts were purified using normal- and reversed-phase flash column chromatography. Final purification of fungal metabolites was carried out by reversed-phase HPLC. Compounds **3**–**7** were structurally characterized based on extensive NMR spectroscopic analyses and MS spectrometric data. Their structures were also confirmed by identical literature data.

Based on the HRESI-MS data, the molecular formula of compound **2** was determined to be C_29_H_44_O. Analysis of the NMR spectra indicated the presence of an ergostane skeleton, with resonances of two adjacent protons at *δ*_H_ 6.21 (d, *J* = 9.7 Hz, 1H) and 5.88 (d, *J* = 9.7 Hz, 1H), suggesting unsaturation in the backbone. These protons, assigned to positions C-7 and C-6, respectively, were confirmed by HMBC correlations (see Figure 2). Additionally, a quaternary ^13^C resonance at *δ*_C_ 149.7, with HMBC cross-peaks to H-7 and H_3_-18, was assigned to C-14. The ergostane skeleton was further revealed to contain unsaturation between C-4 and C-5, as evidenced by the resonance at *δ*_H_ 5.59 (d, *J* = 4.6 Hz, 1H) with COSY correlation with H-3. Furthermore, an additional double bond between positions C-22 and C-23 was supported by the appearance of a characteristic resonance at *δ*_H_ 5.22 (m, 2H) with HSQC cross-peaks to *δ*_C_ 135.4 and 132.1. The presence of a methoxy group at C-3 was confirmed by the ^1^H resonance at *δ*_H_ 3.36 (s, 3H) with an HSQC cross-peak to *δ*_C_ 56.1. The proposed molecular structure of 3-methoxy-ergosta-4,6,8^14^,22-tetraene was deduced from these NMR correlations.

Compound **1** shared the same molecular formula as compound **2**, and the NMR spectra exhibited similar characteristics, indicating an identical planar structure. However, notable differences in the ^1^H and ^13^C NMR resonances at positions 3 and 4 were detected. At position 3 in compound **1**, *δ*_H_ 3.93 and *δ*_C_ 76.6 were observed, whereas in compound **2**, these values shifted to *δ*_H_ 3.73 and to *δ*_C_ 72.5. Similarly, at position 4, compound **1** exhibited resonances at *δ*_H_ 5.49 and *δ*_C_ 123.1, while in compound **2,** resonances *δ*_H_ 5.59 and *δ*_C_ 120.9 were assigned to the same position. The coupling patterns of H-3 in compounds **1** and **2** were also indicative of the different stereochemistry at C-3. While the H-3 proton in compound **1** exhibited a large ^3^*J*_H3-H2_ value of 10.1 Hz (characteristic for an “axial-axial” coupling), the same ^3^*J*_H3-H2_ value in compound **2** was found to be 4.2 Hz (see Table 1). These disparities in the NMR properties strongly suggest a distinct orientation of the methoxy substituent at position C-3.

Based on the NOE correlations, the recommendation is made that compounds **1** and **2** can be identified as C-3 epimers, with compound **1** showing correlations from H_3_-19 to H-1β and H-2β along with H-3 to H-1α, while compound **2** exhibited correlations from H_3_-19 to H-1β and H-2β together with H-3 to H-1β (Figure 3). Although compound **2** displayed identical NMR resonance assignments with that of the natural compound isolated by Lee et al., our detailed stereochemical examination led to the proposal of the α orientation of the OMe group, contrary to their representation [14]. To our knowledge, this research marks the inaugural instance in which both C-3 epimers of the compound were successfully isolated from a natural source, followed by the determination of their stereochemical properties through a comprehensive NMR analysis. Thus, compound **2** was assigned as (3α,22E)-3-methoxy-ergosta-4,6,8^14^,22-tetraene, and compound **1** was characterized as (3β,22E)-3-methoxy-ergosta-4,6,8^14^,22-tetraene. Despite the characterization of the synthetic desmethoxy derivative of compound **2** by Mahé et al. in 1981, the structure has not been reported in the literature [15]. Complete ^1^H and ^13^C NMR resonance assignments for compounds **1** and **2** are provided in Table 1.

Known compounds **4**–**7** were also found to be ergostane-type triterpenes, identified previously from several fungal species. Their NMR and HRMS spectra suggested ergosta-4,6-8,22-tetraen-3-one (**4**) [16,17], ergosta-7,22-dien-3-ol (**5**) [18], 9,11-dehydroergosterol peroxide (**6**) [19,20], and ergosterol peroxide (**7**) [19,20] structures.

Compound **3** was identified as a benzohydrofuran fomannoxin, based on its NMR spectra, HRMS data, and previously published characteristics [21]. Fomannoxin was isolated for the first time from *Fomes annosum* (current name *Heterobasidion annosum*), one of the most important pathogens of coniferous forests, widespread in the Northern Hemisphere [22]. It was also identified in other species, e.g., the termite nest-derived medicinal fungus *Xylaria nigripes* [23], and *Lauriliella taxodii* [24].

The measured NMR and MS spectra for all the isolated compounds (Appendix A). and the complete ^1^H and ^13^C NMR resonance assignments (Appendix A) can be found in the Appendix A. 

Chloroform, *n*-hexane, and methanol extracts of *X. subpileatus* as well as isolated compounds **2**–**7** were further examined for their potential pharmacological properties in tyrosinase, acetylcholinesterase, and butyrylcholinesterase inhibitory assays.

Based on the results obtained from tyrosinase activity experiments (Table 2), fomannoxin (**3**) possesses a considerable inhibitory activity, at 51.62%, while the other isolated fungal constituents have no such activity. It has been proven that tyrosinase is present in the subtantia nigra brain region [25]. As a result of the process involving tyrosinase activity, the level of ROS in this brain area may increase. A higher ROS concentration may lead to increased risk of developing Parkinson’s disease [26]. Our research results indicate that fomannoxin has a high ability to inhibit tyrosinase; hence, it may have an important role in the prevention of Parkinson’s disease. Comparing the acetyl- and butyrylcholinesterase activity test results, one can observe that in both cholinesterase experiments, the chloroform extracts proved to be the most effective, followed by the methanol one, while the *n*-hexane extract provided the lowest activity (Table 3). Furthermore, the AChE enzyme seems to have less susceptibility to the fungal isolates in our assays than the BChE, since all the investigated metabolites except compound **2** exerted some inhibitory activity against BChE, while only compounds **3**–**4** showed activity for AChE. In the case of the tyrosinase assay, here, again, fomannoxin proved to be the most active metabolite, with a notable activity of 67.66, and 83.86% in AChE and BChE assays, respectively. It is known that AChE and BChE are enzymes whose activity is related to the degradation of acetylcholine in the brain. In Alzheimer’s disease, the level of this important neurotransmitter is decreased because of the high expression of cholinesterases [27]. For this reason, cholinesterase inhibitors including fomannoxin are regarded as potential candidates for the development of novel drugs in the therapy of Alzheimer’s disease. Moreover, our cholinesterase inhibitory results for fomannoxin may be considered as a valuable addition to the outcome of a previous study where fomannoxin proved to be the potent neuroprotective compound of the Andean-Patagonian fungi *Aleurodiscus vitellinus* (Lév.) Pat. on a cellular model of amyloid-β peptide toxicity, suggesting a potential anti-Alzheimer disease activity of this metabolite [28]. Fomannoxin, a simple benzofuran structure with a significant biological activity, has the potential to serve as a leading compound for further pharmacological experiments.

## 3. Materials and Methods

### 3.1. General Experimental Procedures

The optical rotations were determined using a Jasco P-2000 digital polarimeter (JASCO International, Co., Ltd., Hachioji, Tokyo, Japan) at the Na_D_ line. The structure elucidation was completed using high-resolution mass-spectrometry techniques: Dionex Ultimate 3000 UHPLC system, which is composed of a 3000RS diode array detector, a TCC-3000RS column thermostat, an HPG-3400RS pump, a SRD-3400 solvent rack degasser, and a WPS-3000TRS autosampler. This system was connected to an Orbitrap Q Exactive Focus Mass Spectrometer possessing an electrospray ionization source (Thermo Fischer Scientific, Waltham, MA, USA). The ionization source was operated both in positive and negative ionization mode, and operation parameter optimization was automatic, working with built-in software. The following experimental parameters were used: spray voltage (+), 3500 V, spray voltage (−), 2500 V; capillary temperature, 320 °C; sheath gas (N2), 47.5 °C; auxiliary gas (N2), 11.25; pare gas (N2), 2.25 arbitrary units. The full scan resolution value was set to 70,000, while the scanning range was established in the range of 120 and 2000 *m*/*z* units. Fragmentation of parent ions formed was performed, applying a normalized collision energy of 15%, 30%, and 45%. VDIA isolation range selection was established based on previous experiments. The fungal samples were dissolved in methanol and filtered through MF-Millipore membrane filters (0.45 μm, mixed cellulose esters) (Billerica, MA, USA).

Flash chromatography (FC) was performed on a CombiFlash Rf+ Lumen instrument (Teledyne Isco, Lincoln, NE, USA) equipped with UV and UV–Vis detection. Normal-phase (silica 80, 40, and 20 g, 0.045–0.063 mm, Molar Chemicals, and RediSep Rf Gold C18, Teledyne Isco, Lincoln, NE, USA) and reversed-phase flash columns (30, 50, and 150 g RediSep Rf Gold C18, Teledyne Isco, Lincoln, NE, USA) were applied as stationary phases in the present experiment. Reversed-phase HPLC purification steps were performed on a Waters 2690 HPLC system, provided by a Waters 996 diode array detector (Waters Corporation, Milford, MA, USA), utilizing a Kinetex C18 100 Å (150 × 10 mm i.d., 5 µm; Phenomenex Inc., Torrance, CA, USA) column. The chemicals applied in chromatographic separations were supplied by Sigma-Aldrich Kft. (Budapest, Hungary) and Molar Chemicals (Halásztelek, Hungary).

NMR spectra were acquired using deuterated chloroform (chloroform-*d*, 99.8 atom% D, with 0.03% (*v*/*v*) tetramethylsilane (TMS), Sigma-Aldrich, Steinheim, Germany) or tetrahydrofuran (tetrahydrofuran-*d*_8_, 99.5 atom% D, Sigma-Aldrich) on a Bruker Avance III HD 600 (600/150 MHz) instrument equipped with a Prodigy cryo-probehead at 295 K. Pulse programs were selected from the Bruker software library (TopSpin 3.5, pl 7). ^1^H and ^13^C chemical shifts (*δ*) are reported in ppm relative to the NMR solvent or the internal standard (TMS). Coupling constants (*J*) are expressed in hertz (Hz). Comprehensive assignments for both ^1^H and ^13^C were achieved using established approaches based on 2D NMR experiments, including ^1^H-^1^H COSY, ^1^H-^1^H ROESY, ^1^H-^13^C HSQC, and ^1^H-^13^C HMBC.

### 3.2. Mushroom Material

Basidiomes of *X. subpileatus* were collected in the Vértes Mountains, Hungary on 6 August 2020, and authenticated by one of the authors (Viktor Papp), based on macro- and micromorphological examinations. The microscopic features of the examined basidiomata were studied in slide preparations mounted in Melzer’s reagent. These sections were observed at a magnification of 1000× through a Zeiss Axio Imager A2 light microscope (Zeiss, Göttingen, Germany) equipped with an attached AxioCam HRc camera (Zeiss, Göttingen, Germany). Axio Vision Release 4.8 software was employed for conducting the measurements. The collected mushroom samples were cleaned of any pollution including soil contaminants and plant parts, then stored at −20 °C. A voucher specimen (No. VPapp-200806xs) was deposited at the Department of Botany, Hungarian University of Agriculture and Life Sciences, Hungary.

### 3.3. Extraction and Isolation

Basidioms of *X. subpileatus* (2.1 kg) were subjected to freeze-drying, which resulted in 275 g of fungal sample. The ground dried sporocarps were extracted with MeOH (14 L) at ambient temperature. The methanol extract (21.17 g) was concentrated under vacuum and then dissolved in 50% aqueous MeOH (600 mL), which was subjected to liquid−liquid extraction, applying the following organic solvents: *n*-hexane (3 × 300 mL), CHCl_3_ (3 × 300 mL), and then EtOAc (3 × 300 mL). For the separation of *n*-hexane fraction (5.28 g), we first performed normal-phase flash chromatography (NP-FC) applying a gradient system of *n*-hexane and acetone (0–35% *n*-hexane gradient elution); then, the similar fractions were combined according to thin-layer chromatography (TLC) (XH1–XH6). Fraction XH4 (530 mg) was further purified by reversed-phase high-performance liquid chromatography (RP-HPLC) applying a mobile phase of water/methanol (75 to 100% gradient elution) to give compounds **1** (5.4 mg) and **2** (2.5 mg). Combined fraction XH3 (352 mg) was purified by RP-HPLC (mobile phase of water/methanol, 75–85% gradient elution), leading to compounds **3** (10 mg) and **4** (21.8 mg). Compounds **6** (6.9 mg) and **7** (33.3 mg) were isolated from fraction XH6 (150 mg) by RP-HPLC (mobile phase of water/methanol, 75 to 100% gradient elution). The chloroform fraction (7.3 g) was further purified by NP-FC, using a gradient system of *n*-hexane and acetone (0–45% *n*-hexane gradient elution). Separation fractions with similar compositions were combined according to TLC monitoring (XC1–XC7). The selected fraction of XC2 (267 mg) was first separated by reversed-phased FC (mobile phase of water/methanol, 60–100% MeOH gradient elution) and finally purified by RP-HPLC (mobile phase/water-methanol, 85 to 95% gradient elution) to give compound **5** (4.8 mg). 

3β-methoxy-ergosta-6,8^14^,22-trien (1): amorphous solid; $\alpha_{D}^{25}$ + 87.8 (*c* 0.42, CHCl_3_, 25.4 °C); HRESIMS *m*/*z* 377.31943 [M + H − CH_3_OH]^+^ (Δ2.2 ppm; C_28_H_41_); HRESI-MSMS (CID = 15%, 30%, 45%; rel int %) *m*/*z* 293, 251.

3α-methoxy-ergosta-6,8^14^,22-trien (2): amorphous solid; $\alpha_{D}^{25}$ − 10.0 (*c* 0.09, CHCl_3_, 25.5 °C) HRESIMS *m*/*z* 377.3192 [M + H − CH_3_OH] + (Δ2.2 ppm; C_28_H_41_); ^1^H and ^13^C NMR data see Table 1; HRESI-MSMS (CID = 15%, 30%, 45%; rel int %) *m*/*z* 293, 251.

### 3.4. Anti-Tyrosinase Activity

The methanol, *n*-hexane, and chloroform extracts (for preparation, see Section 3.3) as well as isolated compounds (**2**–**7**) (for isolation, see Section 3.3) were dissolved in DMSO to obtain a concentration of 8 mg/mL. The spectrophotometric method by Lim et al. [29] was used with some modifications described previously by Studzińska-Sroka et al. [30]. Briefly, 25 µL of the sample, 75 µL of 0.02 M phosphate buffer (pH 6.8), and 50 µL of tyrosinase solution (192 U/mL in phosphate buffer) were mixed. Next, the samples were incubated at room temperature (25 °C) for 10 min with shaking (500 rpm). Subsequently, 50 µL of L-DOPA (2 mM in phosphate buffer) was added and incubated for 20 min with shaking (500 rpm) at the same temperature condition (25 °C). The blanks of samples were prepared using 50 µL of the buffer instead of L-DOPA solutions. The control sample contained DMSO instead of the tested substances. The control blank contained 25 µL of DMSO instead of samples and 50 µL of the buffer instead of L-DOPA solution. The azelaic acid solution was used as the reference. Absorbance was measured at 475 nm (Multiskan GO 1510, Thermo Fisher Scientific, Vantaa, Finland). Two independent experiments were carried out for the investigated substances, and the average from *n* = 2 measurements was calculated. The percentage of tyrosinase inhibition was calculated as follows:$$\mathrm{Tyrosinase}\;\mathrm{inhibition}\;\left[\%\right]=100-\frac{\left(\mathrm{As}\;-\;\mathrm{Abs}\right)}{\left(\mathrm{Ac}\;-\;\mathrm{Abc}\right)}\times 100$$
where As is the absorbance of the sample, Abs is the absorbance of the blank of the sample, Ac is the absorbance of the control, and Abc is the blank of the control. All chemicals used in the tyrosinase activity experiment were from Sigma–Aldrich (St. Louis, MO, USA).

### 3.5. Acetylcholinesterase (AChE) and Butyrylcholinesterase (BChE) Inhibitory Activity

The methanol, *n*-hexane, and chloroform extracts (preparation see Section 3.3) and compounds (**2**–**7**) (for isolation, see Section 3.3) were dissolved in DMSO to obtain a 20 mg/mL concentration. Ellman’s spectrophotometric method [31] was used with some modifications described previously by Studzińska-Sroka et al. [32]. Briefly, 5.0 µL of the sample, 60.0 µL of TRIS-HCl buffer (50 mM, pH = 8), and 30 µL of AChE or BChE (0.2 U/mL) were mixed. Subsequently, the plate was incubated for 5 min at 25 °C with shaking (500 rpm). Next, 30.0 µL of acetylthiocholine iodide (1.5 mM) and 125.0 µL of 5,5′-dithiobis(2-nitrobenzoic acid) (0.3 mM with 10 mM NaCl and 2 mM MgCl_2_·6H_2_O) were added and incubated with shaking (500 rpm) at the same temperature condition (25 °C, 30 min). The blanks of samples were prepared with 30 µL of the buffer instead of AChE. The control sample contained DMSO instead of the test substance. The blank of control contained 5 µL of DMSO instead of samples and 30 µL of the buffer instead of AChE or BChE solution. Absorbance was measured at 405 nm (Multiskan GO 1510, Thermo Fisher Scientific, Vantaa, Finland). Two independent experiments were carried out for the investigated substances, and the average from *n* = 3 measurements was calculated. The percentage of AChE/BChE inhibition was calculated as follows: $$\mathrm{AChE}/\mathrm{BChE}\;\mathrm{inhibition}\;\left[\%\right]=100-\frac{\left(\mathrm{As}\;-\;\mathrm{Abs}\right)}{\left(\mathrm{Ac}\;-\;\mathrm{Abc}\right)}\times 100$$
where As is the absorbance of the sample, Abs is the absorbance of the blank of the sample, Ac is the absorbance of the control, and Abc is the blank of the control. The chemicals used in the cholinesterase activity assays were purchased from Sigma–Aldrich (St. Louis, MO, USA).

### 3.6. Statistical Analysis

The obtained data were expressed as the means ± SE. Statistical analysis was performed using a one-way analysis of variance (ANOVA), and statistical differences (using Duncan’s post hoc tests) with a significance threshold of *p* 0.05 were determined. All statistical analyses were performed using Statistica 13.1 software (StatSoft, Krakow, Poland).

## 4. Conclusions

This study is the first report on the chemical profile of the sporocarps of the wild-growing *X. subpileatus*, providing a thorough study on the isolation and structure determination of the most characteristic biologically active constituents of this crust fungus. Detailed mycochemical examination of the methanol extract of *X. subpileatus* resulted in the identification of seven secondary metabolites, including a new triterpene (**1**), along with a revised (**2**) and four known ergostane constituents **4**–**7**, as well as a benzofuran compound, fomannoxin (**3**). The performed bioactivity experiments highlighted that compounds **3** and **4** exhibited significant inhibitory activity in both AChE and BChE assays, while fomannoxin (**3**) also exerted considerable inhibitory property in tyrosinase experiments. Fomannoxin (**3**), a small benzohydrofuran compound with significant biological activities, shows promise for further pharmacological experiments, especially towards the development of drugs for use in neurodegenerative diseases.

## Figures and Tables

**Figure 1 molecules-29-00213-f001:**
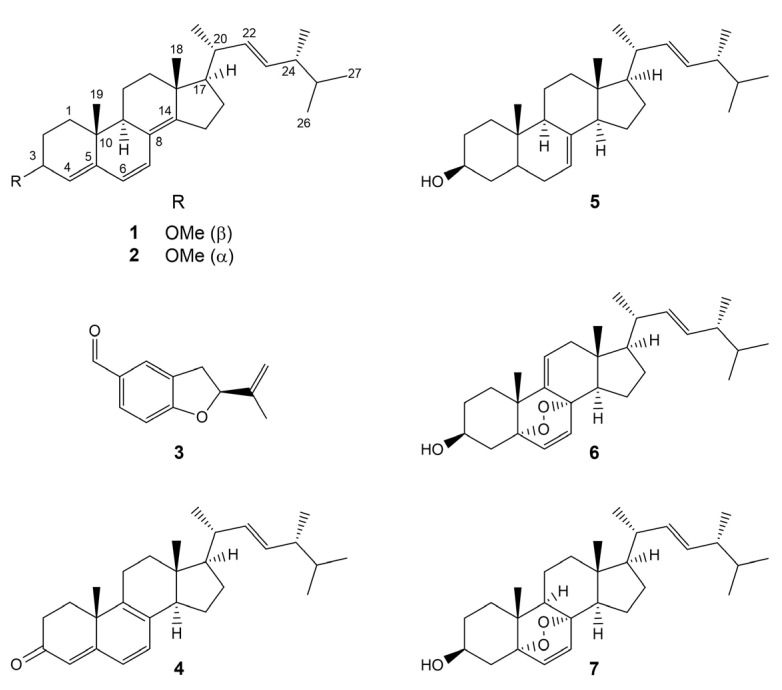
Compounds **1**–**7** isolated from *Xylobolus subpileatus*.

**Figure 2 molecules-29-00213-f002:**
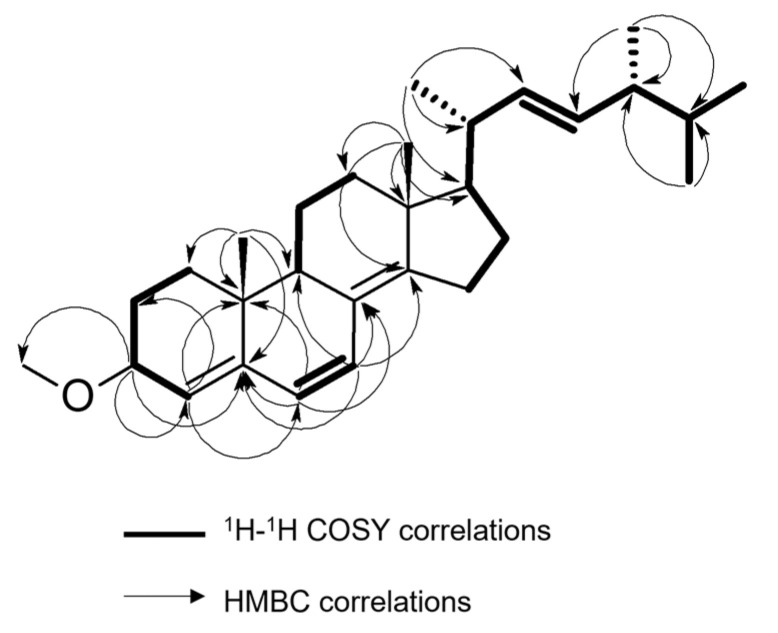
Key ^1^H—^1^H COSY and HMBC correlations in both **1** and **2**.

**Figure 3 molecules-29-00213-f003:**
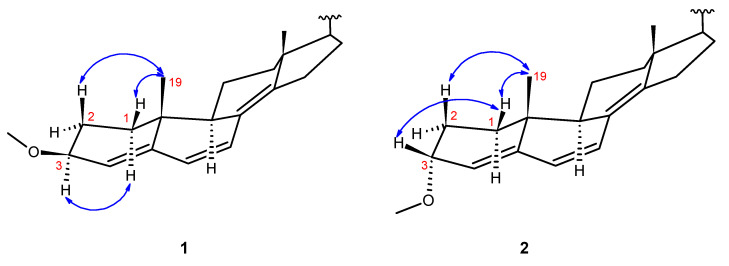
Key ^1^H—^1^H NOE correlations in **1** and **2**.

**Table 1 molecules-29-00213-t001:** Complete ^1^H and ^13^C NMR resonance assignments for compounds **1** and **2**.

No.	1			2		
	*δ* ^13^C	*δ* ^1^H	m, *J*	*δ* ^13^C	*δ* ^1^H	m, *J*
1	33.4	1.73	m	29.7	1.53	m
		1.38	m		1.26	m
2	25.0	2.09	m	23.6	1.96	m
		1.62	m		1.76	m
3	76.6	3.93	ddd, 10.1, 6.5, 2.0 Hz	72.5	3.73	b t, 4.6 Hz
4	123.1	5.49	b s	120.9	5.59	d, 4.6 Hz
5	145.5	-	-	147.4	-	-
6	125.8	5.88	d, 9.6 Hz	125.9	5.88	d, 9.7 Hz
7	126.0	6.17	d, 9.6 Hz	126.8	6.21	d, 9.7 Hz
8	124.8	-	-	124.9	-	-
9	45.5	1.97	m	44.9	2.05	m
10	35.9	-	-	36.0	-	-
11	19.1	1.60	m	19.4	1.64	m
		1.52	m		1.52	m
12	36.3	2.02	m	36.4	2.00	m
		1.26	m		1.25	m
13	43.5	-	-	43.6	-	-
14	149.7	-	-	149.7	-	-
15	25.0	2.40	m	25.0	2.40	m
		2.29	m		2.28	m
16	27.9	1.77	m	27.9	1.77	m
		1.44	m		1.44	m
17	55.9	1.21	m	55.8	1.22	m
18	19.2	0.92	s	19.1	0.92	s
19	18.2	0.89	s	17.1	0.82	s
20	39.4	2.11	m	39.4	2.11	m
21	21.2	1.04	d, 6.7 Hz	21.2	1.04	d, 6.8 Hz
22	135.3	5.21	m	135.4	5.22	m
23	132.2	5.22	m	132.1	5.22	m
24	42.8	1.86	m	42.8	1.87	m
25	33.1	1.48	m	33.1	1.47	m
26	19.7	0.83	d, 6.8 Hz	19.7	0.83	d, 6.8 Hz
27	20.0	0.84	d, 6.8 Hz	20.0	0.84	d, 6.8 Hz
28	17.6	0.92	d, 6.8 Hz	17.6	0.93	d, 6.8 Hz
3-OCH_3_	55.4	3.40	s	56.1	3.36	s

**Table 2 molecules-29-00213-t002:** Anti-tyrosinase activity of extracts and compounds of *Xylobolus subpileatus*.

Compound/ Extract (8 mg/mL)	Inhibition (%)	SD
Methanol extract	25.41 ^c/d^	7.08
Hexane extract	15.51 ^d^	2.88
Chloroform extract	38.33 ^c^	1.16
**2**	na	-
**3**	51.62 ^b^	11.80
**4**	na	-
**5**	na	-
**6**	na	-
**7**	na	-
Azelaic acid (2 mg/mL) *	91.63 ^a^	0.58

Mean values within a column with the same letter are not significantly different at *p* < 0.05 using Duncan’s test. The first letter of the alphabet for the highest values, the next for statistically significant decreasing values. * Reference compound, na: not active.

**Table 3 molecules-29-00213-t003:** Anti-AChE and anti-BChE activities of extracts and compounds of *Xylobolus subpileatus*.

Compound/ Extract (20 mg/mL)	AChE Inhibition (%)	SD	BChE Inhibition (%)	SD
Methanol extract	31.67 ^d^	0.54	45.93 ^b^	3.24
Hexane extract	na	-	22.18 ^c/d^	3.57
Chloroform extract	94.05 ^a^	6.44	86.75 ^a^	9.07
**2**	na	-	na	-
**3**	67.66 ^c^	5.92	83.86 ^a^	9.79
**4**	31.28 ^d^	5.08	32.99 ^c^	14.08
**5**	na	-	6.98 ^e/f^	1.24
**6**	na	-	18.28 ^d/e^	1.38
**7**	na	-	0.49 ^f^	1.67
Galanthamine (0.2 mg/mL) *	76.21 ^b^	1.57	58.38 ^b^	7.21

Mean values within a column with the same letter are not significantly different at *p* < 0.05 using Duncan’s test. The first letter of the alphabet for the highest values, the next for statistically significant decreasing values. * Reference compound, na: not active.

## Data Availability

The data presented in this study are available on request from the corresponding author. The data are not publicly available due to methodological reason.

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
