# Peer review of "Anticholinesterase and Antityrosinase Secondary Metabolites from the Fungus Xylobolus subpileatus"

_molecules, 2023, doi:10.3390/molecules29010213_

Round 1

Reviewer 1 Report

Comments and Suggestions for Authors

The manuscript entitled “Anticholinesterase and antityrosinase secondary metabolites from the fungus Xylobolus subpileatus” is an excellent article where authors reported on the chemical profile of basidiome of the wild-growing X. subpileatus for the first time. It is a very interesting, informative, and well-prepared paper. The writing style and the clarity of the exposition are fine, and the conclusions are clear. I have no specific scientific concern for this paper. In my point of view this paper can be accepted.

1.     What is the significance of the newly identified natural product (compound 1) in X. subpileatus, and how does it broaden our knowledge of the bioactive constituents of this fungus?

2.     Based on the results obtained, what are the potential avenues for future research, and are there specific aspects that warrant further investigation, such as additional bioactivity assays, mechanistic studies, or clinical implications?

3.     Describe the flash chromatography (FC) and reversed-phase high-performance liquid chromatography (RP-HPLC) procedures used for the purification of compounds. What specific conditions and columns were employed in these chromatographic separations?

4.     How was the authenticity of the collected mushroom material ensured?

5.     How were the extracts and compounds prepared, and what were the conditions of the spectrophotometric assays?

6.     How were the samples prepared, and what were the conditions for measuring tyrosinase inhibition?

7.     Given the notable inhibitory activity of fomannoxin, especially in cholinesterase assays, what potential biological relevance does it hold, and how could these findings contribute to further studies?

Author Response

Thank you for the helpful and valuable comments you made, we really appreciate the time devoted to reviewing our manuscript.

Please find below our detailed responses to the questions raised:

Reviewer 1

  1. What is the significance of the newly identified natural product (compound 1) in X. subpileatus, and how does it broaden our knowledge of the bioactive constituents of this fungus?

Fungi are prolific synthesizers of natural products of astonishing structural diversity possessing a wide range of biological activities. Among them triterpene constituents represent one of the most characteristic classes of secondary metabolites. Members of this vast group of compounds including the above-mentioned new compound from Xylobolus subpileatus are known to possess cytotoxic, MDR reversal activity (e.g. Yazdani et al. Triterpenes from Pholiota populnea as Cytotoxic Agents and Chemosensitizers to Overcome Multidrug Resistance of Cancer Cells. Journal of Natural Products, 2021), and anti-inflammatory properties (e.g. Yazdani et al. Pholiols E–K, lanostane-type triterpenes from Pholiota populnea with anti-inflammatory properties. Phytochemistry, 2022), among others. Considering the ethnomedicinal potential of this fungus it would be interesting to assess the contribution of these triterpenoid compounds to the potential pharmacological activities of X. subpileatus.

  1. Based on the results obtained, what are the potential avenues for future research, and are there specific aspects that warrant further investigation, such as additional bioactivity assays, mechanistic studies, or clinical implications?

Thank you for the question raised. One of the isolated compounds, namely fomannoxin exerted remarkable activity in both cholinesterase and tyrosinase inhibitory the assays. This small molecule compound worths to be investigated in further pharmacological assays. Here we can have multiple possibilities. One option is to perform some studies to reveal the specific mechanisms of action behind the observed pharmacological activities. Other option can be the investigation of several semisynthetic derivatives of fomannoxin in order to identify specific structure-activity relationships. Other important feature to assess would be the toxicity profile. If the results of the above-mentioned assays are sound then animal experiments can follow, which finally can pave the way for potential clinical investigations.

  1. Describe the flash chromatography (FC) and reversed-phase high-performance liquid chromatography (RP-HPLC) procedures used for the purification of compounds. What specific conditions and columns were employed in these chromatographic separations?

Flash chromatography was carried out on a CombiFlash Rf+ Lumen instrument (Teledyne Isco, Lincoln, NE, USA) equipped with UV and UV–Vis detection. Normal phase (silica 80, 40, and 20 g, 0.045–0.063 mm, Molar Chemicals, and RediSep Rf Gold C18, Teledyne Isco, Lincoln NE, USA) and reversed phase flash columns (30, 50, and 150 g RediSep Rf Gold C18, Teledyne Isco, Lincoln NE, USA) were applied as stationary phases in the present experiment. As mobile phases we applied n-hexane-acetone (normal phase separations) and water-methanol systems (reversed phase separations). Reversed phase HPLC purification steps were performed on a Waters 2690 HPLC system provided with a Waters 996 diode array detector (Waters Corporation, Milford, MA, USA) utilizing a reversed phase Kinetex C18 100 Å (150 × 10 mm i.d., 5 µm; Phenomenex Inc, Torrance, CA, USA) column. As mobile phase water-methanol elution system was applied. The chemicals applied in chromatographic separations were supplied by Sigma-Aldrich Kft. (Budapest, Hungary) and Molar Chemicals (Halásztelek, Hungary).

  1. How was the authenticity of the collected mushroom material ensured?

The microscopic features of the examined basidiomata were studied in slide preparations mounted in Melzer's reagent. These sections were observed at a magnification of 1000× through a Zeiss Axio Imager A2 light microscope (Zeiss, Göttingen, Germany) equipped with an attached AxioCam HRc camera (Zeiss, Göttingen, Germany). Axio Vision Release 4.8 software was employed for conducting the measurements.

  1. How were the extracts and compounds prepared, and what were the conditions of the spectrophotometric assays?

and

  1. How were the samples prepared, and what were the conditions for measuring tyrosinase inhibition?

The methodology used for enzyme inhibition assay are defined in sections 3.4. and 3.5 of the manuscript. We have described the preparation of samples used for enzyme inhibition analysis (extracts or compounds dissolved in DMSO) and the spectrophotometric parameters (type of spectrophotometer, length of wave, and temperature of experiments). However, we have added information where (in the manuscript) the preparation of extracts and isolation of compounds is described (point 3.3).

3.4 Anti-tyrosinase activity

The methanol, hexane, chloroform extracts (preparation see 3.3) and compounds (1-7) (isolation see 3.3) were dissolved in DMSO to obtain a concentration of 8 mg/mL. The spectrophotometric method by Lim et al. [26] was used with some modifications described previously by StudziÅ„ska-Sroka et al. [27]. Briefly, 25 µL of the sample, 75 µL of 0.02 M phosphate buffer (pH 6.8), and 50 µL of tyrosinase solution (192 U/mL in phosphate buffer) were mixed. Next, the samples were incubated at room temperature (25 °C) for 10 min with shaking (500 rpm). Subsequently, 50 µL of L-DOPA (2 mM in phosphate buffer) was added and incubated for 20 min with shaking (500 rpm) at the same temperature condition (25 °C). The blanks of samples were prepared using 50 µL of the buffer instead of L-DOPA solutions. The control sample contained DMSO instead of the tested substances. The control blank contained 25 µL of DMSO instead of samples and 50 µL of the buffer instead of L-DOPA solution. The azelaic acid solution was used as the reference. Absorbance was measured at 475 nm (Multiskan GO 1510, Thermo Fisher Scientific, Vantaa, Finland). Two independent experiments were carried out for the investigated substances, and the average from n = 2 measurements was calculated. The percentage of tyrosinase inhibition was calculated as follows:

Tyrosinase inhibition [%]=100-  ((As – Abs))/((Ac – Abc))×100

where: As is the absorbance of the sample, Abs is the absorbance of the blank of the sample, Ac is the absorbance of the control and Abc is the blank of the control. All chemicals used in the tyrosinase activity experiment were from Sigma–Aldrich (St. Louis, CA, USA).

3.5 Acetylcholinesterase (AChE)- and butyrylcholinesterase (BChE) inhibitory activity

The methanol, hexane, chloroform extracts (preparation see 3.3) and compounds (1-7) (isolation see 3.3)  were dissolved in DMSO to obtain a 20 mg/mL con-centration. Ellman’s spectrophotometric method [28] was used with some modifications described previously by StudziÅ„ska-Sroka et al. [29]. Briefly, 5.0 µL of the sample, 60.0 µL of TRIS-HCl buffer (50 mM, pH = 8), and 30 µL of AChE or BChE (0.2 U/mL) were mixed. Subsequently, the plate was incubated for 5 min at 25 °C with shaking (500 rpm). Next, 30.0 µL of acetylthiocholine iodide (1.5 mM) and 125.0 µL of 5,5′-dithiobis(2-nitrobenzoic acid) (0.3 mM with 10 mM NaCl and 2 mM MgCl2·6H2O) were added and incubated with shaking (500 rpm) at the same temperature condition (25 °C, 30 min). The blanks of sam-ples were prepared with 30 µL of the buffer instead of AChE. The control sample con-tained DMSO instead of the test substance. The blank of control contained 5 µL of DMSO instead of samples and 30 µL of the buffer instead of AChE or BChE solution. Absorb-ance was measured at 405 nm (Multiskan GO 1510, Thermo Fisher Scientific, Vantaa, Finland). Two independent experiments were carried out for the investigated substances, and the average from n = 3 measurements was calculated. The percentage of AChE/BChE inhibition was calculated as follows:

AChE/BChE inhibition [%]=100- ((As – Abs))/((Ac – Abc))×100

where: As is the absorbance of the sample, Abs is the absorbance of the blank of the sample, Ac is the absorbance of the control and Abc is the blank of the control. The chemicals used in the cholinesterase activity assays were purchased from Sigma–Aldrich (St. Louis, CA, USA).

  1. Given the notable inhibitory activity of fomannoxin, especially in cholinesterase assays, what potential biological relevance does it hold, and how could these findings contribute to further studies?

The appropriate amendments have been added to the results and discussion section of the manuscript.

For tyrosinase assay we have added the following:

“It has been proven that tyrosinase is present in the brain region called subtantia nigra [Carballo-Carbajal et al]. As a result of the process involving tyrosinase activity, the level of ROS in this brain area may increase. Higher ROS concentration may lead to increased risk of developing Parkinson's disease [Dias et al.]. Our research results indicate that fomannoxin has a high ability to inhibit tyrosinase; hence, it may have an important role in the prevention of Parkinson's disease.”

  • Carballo-Carbajal, I., Laguna, A., Romero-Giménez, J. et al. Brain tyrosinase overexpression implicates age-dependent neuromelanin production in Parkinson’s disease pathogenesis. Nat Commun 10, 973 (2019). https://doi.org/10.1038/s41467-019-08858-y
  • Dias V, Junn E, Mouradian MM. The role of oxidative stress in Parkinson's disease. J Parkinsons Dis. 2013;3(4):461-91. doi: 10.3233/JPD-130230. PMID: 24252804; PMCID: PMC4135313.

For AChE and BChE analysis we have added the following:

“It is known that AChE and BChE are the enzymes whose activity is related to the deg-radation of acetylcholine in the brain. In Alzheimer's disease, the level of this im-portant neurotransmitter is decreased because of the high expression of cholinesterases [Sacks et al.]. For this reason, cholinesterase inhibitors including fomannoxin are regarded as potential candidates for the development of novel drugs in the therapy of Alzheimer's disease. Moreover, our …”

•             Sacks D, Baxter B, et al. Multisociety Consensus Quality Improvement Revised Consensus Statement for Endovascular Therapy of Acute Ischemic Stroke. International Journal of Stroke. 2018;13(6):612-632. doi:10.1177/1747493018778713

Reviewer 2 Report

Comments and Suggestions for Authors

Reviewer’s Comments on Manuscript, Molecules-2747628-Peer-Review-V1

The article entitled “Anticholinesterase and Antityrosinase Secondary Metabolites from the Fungus Xylobolus subpileatus” is a study based on the chemical profile of the sporocarps of the wild growing X. subpileatus, providing an insight analysis of the isolation and structural determination of the biologically active constituents of this crust fungus. Although the manuscript is written well but it has some serious problems relating to the structural elucidation. The major points of concern are as follows.

1.     The structure elucidation has been poorly discussed throughout this manuscript. The stereochemical assignments of the OMe group at C-3 is uncertain in both compound-1 and 2. The given information is insufficient in assigning the stereochemistry of the OMe group at C-3. HMBC interactions are not helpful here in assigning the stereochemistry at this position. Therefore, the statement at Line-106, Page-3 “stating that “compound 1 was found to be the C-3 epimer of compound 2” seems quite premature at this stage.  

2.     Although the NOESY interactions in case of a-OMe (H-3 with H3-19) and its absence in b-OMe could be helpful somehow in assigning stereochemistry but still its confirmation through specific rotation is mandatory here. But we do not see such explanation here. Thus, the claimed compounds’ structure cannot be confirmed.

3.     In all the known compounds we see the same type of issues. Only by simply comparing the proton and 13C-NMR or HMBC and HSQC data with the literature cannot guarantee the accuracy of the claimed structures’ identity unless the NOESY interactions and specific rotation information are not utilized in assigning the stereochemistry.

4.     The authors are encouraged to discuss the stereochemistry in all the positions of all the compounds, how they arrived at the conclusions that the information provided are sufficient enough in claiming the structures of these compounds.

5.     Authors are encouraged to provide COSY, HSQC and HMBC relationships in new compound by using proper arrows on the structure.

6.     In the “extraction and isolation” section n-hexane fraction is supposed to have less polar compounds that can easily be eluted through normal phase chromatography but here reverse phase HPLC technique was used for the purification of compounds from n-hexane sub fractions. Here authors are encouraged to justify the reason.

7.     Statistical tools applied for the calculations of experimental data (of any activity) are not discussed. Authors are encouraged to write data processing approach in section of material and methods. Also provide statistical data regarding significant differences between different groups.

8.     In conclusion section, in Line-80, the term “novel triterpene (1)” is used while compound 1 is claimed in this manuscript as new compound not as novel.

After addressing the above major concerns, the authors are required to resubmit this manuscript.

Author Response

Thank you for the helpful and valuable comments you made, we really appreciate the time devoted to reviewing our manuscript.

Please find below our detailed responses to the questions raised:

  1. The structure elucidation has been poorly discussed throughout this manuscript. The stereochemical assignments of the OMe group at C-3 is uncertain in both compound-1and 2. The given information is insufficient in assigning the stereochemistry of the OMe group at C-3. HMBC interactions are not helpful here in assigning the stereochemistry at this position. Therefore, the statement at Line-106, Page-3 “stating that “compound 1 was found to be the C-3 epimer of compound 2” seems quite premature at this stage.

We have revised the manuscript to provide a more detailed description of the stereochemical structure including a more comprehensive analysis of the available data. It's crucial to clarify that HMBC interactions were not utilized to define the stereochemistry at C-3. We understand the limitations of HMBC in this context and have reframed the relevant sections to avoid any misunderstanding. The statement "compound 1 was found to be the C-3 epimer of compound 2," has been reviewed and modified to reflect a more cautious interpretation. 

  1. Although the NOESY interactions in case of a-OMe (H-3 with H3-19) and its absence in b-OMe could be helpful somehow in assigning stereochemistry but still its confirmation through specific rotation is mandatory here. But we do not see such explanation here. Thus, the claimed compounds’ structure cannot be confirmed.

We acknowledge the importance of the observed NOE interactions, particularly in distinguishing between α-OMe and β-OMe, considering that the chemical shifts differed only for this part of the structure. To supplement these findings, we conducted optical rotation measurements for the new compound, as detailed in the “Extraction and isolation” chapter. The NOESY correlations, combined with the supporting literature data, are considered indicative of the presented stereochemistry. In response to the reviewer's request, we have expanded the discussion to provide a more comprehensive understanding of our methodology.

  1. In all the known compounds we see the same type of issues. Only by simply comparing the proton and 13C-NMR or HMBC and HSQC data with the literature cannot guarantee the accuracy of the claimed structures’ identity unless the NOESY interactions and specific rotation information are not utilized in assigning the stereochemistry.

We would like to emphasize that we agree with the point that solely comparing 1H and 13C-NMR or HMBC and HSQC data with the literature may not suffice to guarantee the accuracy of claimed structures. Our approach included independent experiments involving detailed HRMS and NMR experiments, including various techniques such as 1H, 13C, COSY, HSQC, HMBC, and ROESY. This comprehensive analysis was followed by a complete 1H and 13C NMR assignment to validate the proposed structures. To verify the resulting structures, we cross-referenced them with existing literature. We made a conscious decision not to publish redundant detailed structural data for already well-documented molecules as natural compounds. For transparency, all relevant spectra and tabular NMR data have been included in the supplementary file to support FAIR scientific data principles.

  1. The authors are encouraged to discuss the stereochemistry in all the positions of all the compounds, how they arrived at the conclusions that the information provided are sufficient enough in claiming the structures of these compounds.

In responding to the inquiry regarding the comprehensive discussion of stereochemistry of all chiral centres of the compounds, we posit that, for our present mycochemical research, a detailed examination of stereochemistry at each position may not be imperative. Considering the well-established biosynthetic pathways and the extensive literature on fungal metabolites, we find it adequate to emphasize specific stereochemical details. Our focus is not on challenging the fundamental triterpene structure, as that is beyond the scope and purpose of this research. However, the updated manuscript now includes a more detailed discussion of the definition of the C-3 stereochemical configuration for compounds 1 and 2.

  1. Authors are encouraged to provide COSY, HSQC and HMBC relationships in new compound by using proper arrows on the structure.

We agree with the Reviewer, and we do appreciate the suggestion. In the updated manuscript, we have included COSY and HMBC correlations, appropriately indicated with arrows on the structure of the new compound.

  1. In the “extraction and isolation” sectionn-hexane fraction is supposed to have less polar compounds that can easily be eluted through normal phase chromatography but here reverse phase HPLC technique was used for the purification of compounds from n-hexane sub fractions. Here authors are encouraged to justify the reason.

Thank you for your question. We agree on the fact that hexane fraction contains less polar compounds then the chloroform one, however there could be an overlap in compounds between the two fractions depending on the liquid-liquid partition performed and the fungal metabolites contained. Fungal fruiting bodies are known to synthesize a large variety of triterpene compounds with varying polarities depending on the diverse substituents in different positions of the basic skeleton. Regarding you comment during the separation of the hexane fraction we had access only to a reversed phase HPLC, which however proved to be useful in the successful isolation of the specified compound(s).

  1. Statistical tools applied for the calculations of experimental data (of any activity) are not discussed. Authors are encouraged to write data processing approach in section of material and methods. Also provide statistical data regarding significant differences between different groups.

Thank you for this comment. The statistical analysis has been completed (Table 2 and Table 3), as well as in the text of the manuscript:

“3.6. Statistical Analysis

The obtained data were expressed as the means ± SE. Statistical analysis was per-formed using a one-way analysis of variance (ANOVA), and statistical differences (using Duncan’s post-hoc tests) with a significance threshold of p 0.05 were determined. All sta-tistical analyses were performed using Statistica 13.1 software (StatSoft, Poland)”

  1. In conclusion section, in Line-80, the term “novel triterpene (1)” is used while compound is claimed in this manuscript as new compound not as novel.

Thank you for your comment, “novel triterpene” has been corrected to “new triterpene”.

Reviewer 3 Report

Comments and Suggestions for Authors

Author Response

The article, “Anticholinesterase and antityrosinase secondary metabolites from the fungus Xylobolus subpileatus”, by Kristóf Felegyi et. al., provided a comprehensive study on the isolation and structure determination of biologically active constituents of X.subpileatus fungus. By mycochemical analysis of the methanol extracts, they have identified & isolated seven secondary metabolites. Extraction, Isolation and Purification of these metabolites were clearly written in the manuscript. The measured NMR & MS spectra and their 13C NMR resonance assignments of these seven metabolites were nicely presented in the supplementary material. The authors were also measured anti-tyrosinase activity and also anti-AchE & anti-BChE activites of extracts and compared with compounds of X.subpileatus. The performed bioactivity experiments highlighted that, compounds 3 and 4 exhibited significant inhibitory activity in both AChE and BChE assays (~68% &32% respectively for compound 3, while ~84% & 33% for compound 4), while 3 also exerted considerable inhibitory property in tyrosinase experiments (~51%). Overall, the research article provides a comprehensive report on the chemical profile of the sporocarps of the wild growing X.subpileatus fungus. Interestingly, the compound 3 Fomannoxin, a small benzofuran compound with significant biological activities shows promise for further pharmacological experiments to be addressed.

Thank you for the helpful and valuable comments you made, we really appreciate the time devoted to reviewing our manuscript. In the current manuscript we tried to provide a comprehensive report on the chemical profile of the sporocarps of the wild growing X. subpileatus fungus. We agree with the reviewer that fomannoxin a small benzofuran compound possess significant biological activities and shows promise for further pharmacological assays.

Round 2

Reviewer 2 Report

Comments and Suggestions for Authors

The revised version of the manuscript entitled “Anticholinesterase and Antityrosinase Secondary Metabolites from the Fungus Xylobolus subpileatus has been thoroughly reviewed and was found in sound condition. The authors have thoroughly improved the manuscript according to the review’s suggestions. The HMBC interactions and NOE correlations have been properly indicated in the revised version. The authors have given satisfactory answers to most of the questions raised. Therefore, I recommend this manuscript for publication in this journal in this current form.    

Author Response

Thank you for the helpful and valuable comments you made, we really appreciate the time devoted to reviewing our revised manuscript.